# Evaluating the impact of an intervention to increase uptake of modern contraceptives among adolescent girls (15–19 years) in Nigeria, Ethiopia and Tanzania: the Adolescents 360 quasi-experimental study protocol

Christina Joanne Atchison,[1] Emma Mulhern,[2] Saidi Kapiga,[3,4] Mussa Kelvin Nsanya,[4] Emily E Crawford,[5] Mohammed Mussa,[6] Christian Bottomley,[1] James R Hargreaves,[7] Aoife Margaret Doyle[1]

For numbered affiliations see end of article.

**Correspondence to**
Dr Christina Joanne Atchison; christina.atchison@lshtm.ac.uk

## ABSTRACT

**Introduction** Nigeria, Ethiopia and Tanzania have some of the highest teenage pregnancy rates and lowest rates of modern contraceptive use among adolescents. The transdisciplinary Adolescents 360 (A360) initiative being rolled out across these three countries uses human-centred design to create context-specific multicomponent interventions with the aim of increasing voluntary modern contraceptive use among girls aged 15–19 years.

**Methods** The primary objective of the outcome evaluation is to assess the impact of A360 on the modern contraceptive prevalence rate (mCPR) among sexually active girls aged 15–19 years. A360 targets different subpopulations of adolescent girls in the three countries. In Northern Nigeria and Ethiopia, the study population is married girls aged 15–19 years. In Southern Nigeria, the study population is unmarried girls aged 15–19 years. In Tanzania, both married and unmarried girls aged 15–19 years will be included in the study. In all settings, we will use a prepopulation and postpopulation-based cross-sectional survey design. In Nigeria, the study design will also include a comparison group. A one-stage sampling design will be used in Nigeria and Ethiopia. A two-stage sampling design will be used in Tanzania. Questionnaires will be administered face-to-face by female interviewers aged between 18 and 26 years. Study outcomes will be assessed before the start of A360 implementation in late 2017 and approximately 24 months after implementation in late 2019.

**Ethics and dissemination** Findings of this study will be widely disseminated through workshops, conference presentations, reports, briefings, factsheets and academic publications.

## INTRODUCTION

In 2012, the global community launched the Family Planning 2020 (FP2020) initiative to reach 120 million new contraceptive users in developing countries by 2020.[1]

---

### Strengths and limitations of this study

► This study is part of an independent multicomponent impact evaluation of a multicountry adolescent sexual and reproductive health intervention, and we will collect comparable data before and after intervention implementation in four different settings in three countries.

► In Nigeria, the randomisation of intervention allocation was not possible; therefore, in this quasi-experimental design, conclusions about causality are less definitive than a cluster-randomised design.

► Triangulation with dose–response and trends analyses will strengthen the inference possible from the study findings.

► Due to resource constraints, we focused resources on selected geographical areas, and we acknowledge that this will restrict the generalisability of our findings.

---

Specifically, it calls for meeting all women's needs for modern contraception to prevent unintended pregnancies and reducing the high adolescent birth rates in the world's poorest countries. Helping adolescent girls avoid unintended pregnancies can have far-reaching benefits for them, their children and societies as a whole.[2] Complications of pregnancy and childbirth are the leading cause of death among girls aged 15–19 years,[3 4] and babies born to adolescent mothers face greater health risks than those born to older women.[5] Moreover, adolescent childbearing is associated with lower educational attainment, and it can perpetuate a cycle of poverty from one generation to the next.[3]

From 1990 to 2010, adolescent fertility rates declined in most countries.[6] However, adolescent fertility rates remain high in many lower-income countries. In some countries, fertility rates are declining more slowly in adolescents than in older women.[1] Nigeria, Ethiopia and Tanzania have among the highest rates of adolescent fertility globally, with 109, 57 and 118 births per 1000 girls aged 15–19 per year, respectively.[1] Equally, these countries also have some of the lowest rates of use of modern contraception in adolescents. In Nigeria, 98.8% of married adolescent girls and 50.3% of unmarried sexually active adolescent girls do not use a modern contraceptive method.[7] The equivalent figures are 68.2% and 42.5% in Ethiopia[8] and 86.7% and 66.9% in Tanzania.[9] Preventing pregnancy among adolescents is a global priority and new interventions are needed, particularly in countries like Nigeria, Ethiopia and Tanzania where adolescent fertility rates remain high.

Despite having clear needs, both married and unmarried adolescent girls in many low-income and middle-income countries lack access to reliable contraception. In addition, rates of contraceptive failure are higher in adolescents than in older women, with younger women more likely to abandon contraception despite ongoing need.[10] Reasons include poor understanding of pregnancy risks, concerns about the effect of contraceptives on health or fertility and opposition from partners. Lack of knowledge of services, cost, shyness and community stigma about sexual activity and disapproving attitudes from providers are further barriers.[10] Adolescent sexual and reproductive health is affected by a country's cultural, religious, legal, political and economic contexts. In responding, health actions are needed at each level, from structural, through to community settings including schools and health services.[4 10] The most effective programmes are typically multicomponent and target one or more of these settings.[4 10] There are a range of effective and scalable interventions including comprehensive sexuality education and provision of youth-friendly sexual and reproductive health services. Yet the evidence base for action remains relatively weak.[4 11] The overwhelming majority of intervention studies derive from high-income countries.[11]

Adolescents 360 (A360) is an initiative in the field of adolescent sexual and reproductive health programming, with the intention of being implemented at scale in Nigeria, Ethiopia and Tanzania. The final package of interventions is country-specific and includes a combination of community-based sexual and reproductive health education, counselling and improved contraceptive provision through 'adolescent friendly' services. The external evaluation of the A360 intervention comprises an outcome evaluation, a process evaluation and a cost effectiveness study. We present here the study protocol for the outcome evaluation of A360.

## METHODS AND ANALYSIS
### Study objective
The primary objective of the outcome evaluation is to evaluate the effectiveness of the A360 programme in increasing the uptake of voluntary modern contraception among sexually active girls aged 15–19 years.

### Study settings
In Nigeria, A360 is being implemented by the Society for Family Health (SFH)[12] in three states in the north (Federal Capital Territory, Nasarawa, Kaduna) and in seven states in the south (Lagos, Osun, Ogun, Oyo, Edo, Delta and Akwa Ibom) of the country. The A360 programme will be implemented in approximately 60% of the local government areas (LGAs) in each selected state. We will conduct the outcome evaluation in four LGAs in Nasarawa State and two LGAs in Ogun State. Nasarawa State is a state in north central Nigeria with a total population of 1.9 million.[13] Overall, 41.3% of the female household population have no education, and the median age at first marriage for women is 19.7 years.[7] Current use of modern contraception among married women aged 15–49 years is 16.3%. Ogun State in south western Nigeria has a total population of 3.8 million.[13] Overall, 24.1% of the female household population have no education, and the median age at first marriage for women is 20.5 years. Current use of modern contraception among married women aged 15–49 years is 21.5%.[7]

In Ethiopia, A360 is being implemented by Population Services International (PSI) in two city administrations and five regional states (Addis Ababa, Amhara, Dire Dawa, Harari, Oromia, Southern Nations, Nationalities and People's Region and Tigray). Within each of the selected regional states, A360 will be implemented in selected woredas (districts), and we will conduct the outcome evaluation study in four woredas in Oromia Region. Oromia Region has a total population of 27 million.[14] Overall, 51.5% of the female household population have no education, and the median age at first marriage for women is 17.4 years. Current use of modern contraception among married women aged 15–49 years is 28.1%.[8]

In Tanzania, A360 is being implemented by PSI in 10 regions (Kagera, Geita, Mwanza, Arusha, Tabora, Tanga, Dar es Salaam, Mbeya, Iringa and Morogoro). We will conduct the outcome evaluation in urban and semiurban wards of Ilemela District, Mwanza Region. Mwanza Region has a total population of 2.8 million.[15] Overall, 24.2% of the female household population have no education, and the median age at first marriage for women is 18.9 years. Current use of modern contraception among married women aged 15–49 years is 18.4%.[9]

### Interventions under study
The A360 interventions are being designed using a human-centred design process which includes the following steps[16]:

**Table 1** The likely final A360 package of interventions in each setting

| A360 country | Target population | Intervention |
|---|---|---|
| Nigeria (North) | Married girls | Under design at the time of writing. Due to security issues during the design phase, it was not feasible to develop a context specific programme. A rapid assessment based on insights and prototypes from the other contexts is underway to determine the programme to rollout in Northern Nigeria. |
| Nigeria (South) | Unmarried girls | *'9ja Girls'*<br>► Mobilisation of girls aged 15–19 to attend 9ja Girls events by emphasising vocational skills and life planning.<br>► Sensitisation sessions in the community with mothers.<br>► Community launches involving key community influencers (eg, local government, religious leaders).<br>► Physical (eg, in a public health centre) and digital (eg, online forums) safe spaces for girls.<br>► Public health centre-based vocational skills classes focusing on job skills and life planning, including opt-out one-to-one counselling sessions with adolescent friendly providers to address fears, dispel myths and highlight benefits of contraception. Opt-out means that girls will be counselled by a service provider unless they decline.<br>► Public health centre-based delivery of family planning products and/or referral to adolescent friendly providers.<br>*'9ja Girls'* will use clusters of public health centres, private social-franchise clinics and stand-alone clinics in facilities donated by the Ministry of Health. |
| Ethiopia | Married girls | *'Smart Start'*<br>► Community-based financial planning linked to family planning counselling sessions for newly married or soon to be married couples/girls to enable informed choice and decision making.<br>► Delivered in partnership with the national Health Extension Programme via Health Extension Workers and augmented by the existing community infrastructure of the Women's Development Army and a PSI-recruited *'Smart Start'* team and local Youth Champions.<br>► Delivery of family planning products through local service providers. |
| Tanzania | Unmarried and married | *'Kuwa Mjanja'* (Be Smart)<br>► Mobilisation of girls aged 15–19 to attend *'Kuwa Mjanja'* events by emphasising vocational skills, learning about body changes and/or planning for life goals.<br>► Community and clinic-based events for mothers to sensitise them to their daughters' developmental stages associated with the desire to use contraception to prevent unwanted pregnancy in adolescent girls.<br>► Pop-up and clinic-based events focusing on vocational skills and life planning for girls, including opt-out one-to-one counselling sessions with adolescent friendly providers to address fears, dispel myths and highlight benefits of contraception. Opt out means that girls will be counselled by a service provider unless they decline.<br>► Community-based and clinic-based delivery of family planning products.<br>► Sustained interaction and engagement through *'Kuwa Mjanja'* branded clubs and social media forums.<br>► Club-based events focusing on vocational skills, learning about their bodies, reproductive health and contraception.<br>*'Kuwa Mjanja'* will use both public and private social-franchise clinics in partnerships with the Ministry of Health, NGOs and community-based organisations. |

A360, Adolescents 360; PSI, Population Services International.

1. Inspiration: a period of formative research to understand adolescent girls' sexual and reproductive health needs and their sociocultural environment.
2. Ideation: an iterative process of generating, testing and refining ideas and developing and testing prototypes in real-world settings.
3. Implementation: intervention rollout at scale in target regions across the three countries.

Interventions are currently in the final round of prototyping (ideation phase). The most likely final package of interventions in each setting is described in table 1.

### Design of outcome evaluation

A summary of the methods used can be found in table 2. Separate protocols were developed for each country to take into account the country-specific A360 implementation strategies and study designs developed.

#### Nigeria

The intervention will be evaluated in Ogun (South Nigeria) and Nasarawa (North Nigeria) through population-based surveys conducted at baseline (late 2017) and approximately 24 months after the start of the

**Table 2** Summary of methods

| A360 country | A360 regions | Study design | Outcome evaluation study setting | Study population (sample size) | Sampling strategy |
|---|---|---|---|---|---|
| Nigeria (South) | Lagos, Osun, *Ogun*, Oyo, Edo, Delta, Akwa Ibom | Cross-sectional before-and-after study with comparison group | Ogun State:<br>▲ Ado-Odo Ota LGA (intervention)<br>▲ Shagamu LGA (comparison) | Unmarried girls aged 15–19 years (12 020)<br>Cohabiting adults (250) | Single stage cluster design<br>PSU: EA<br>Simple random sample of EAs (approx. 710 in Ogun). All HHs visited in selected EAs. All eligible individuals to be recruited to participate. |
| Nigeria (North) | Federal Capital Territory, Kaduna, *Nasarawa* | Cross-sectional before-and-after study with comparison group | Nasarawa State:<br>▲ Doma LGA (intervention)<br>▲ Toto LGA (comparison)<br>▲ Karu LGA (intervention)<br>▲ Nasarawa LGA (comparison) | Married girls aged 15–19 years (4555)<br>Husband (250) | Single stage cluster design<br>PSU: EA<br>Simple random sample of EAs (approx. 1150 in Nasarawa). All HHs visited in selected EAs. All eligible individuals to be recruited to participate. |
| Ethiopia | Addis Ababa, Amhara, Dire Dawa, Harari, *Oromia*, SNNP, Tigray | Cross-sectional before-and-after study | Oromia regional state:<br>Wara Jarso, Lome, Ada'a, Fentale woredas | Married girls aged 15–19 years (1926)<br>Husband (128) | Single stage cluster design<br>PSU: Kebele<br>Probability proportional to size sample of 45 kebele. All HHs visited in selected kebele. All eligible individuals to be recruited to participate. |
| Tanzania | Kagera, Geita, *Mwanza*, Arusha, Tabora, Tanga, Dar es Salaam, Mbeya, Iringa, Morogoro | Cross-sectional before-and-after study | Mwanza region:<br>Ilemela district (urban and semiurban wards only) | Married and unmarried girls aged 15–19 years (4980)<br>Cohabiting adults (127) | Two stage cluster design<br>PSU: Street<br>Simple random sample of 30 streets. Simple random sample of 50 GPS points within each street. All HHs with front door within 20m radius of GPS point visited. All eligible individuals recruited to participate. |

A360, Adolescents 360; EA, enumerations area; GPS, Global Positioning System; HH, household; LGA, local government area; PSU, primary sampling unit; SNNP, Southern Nations, Nationalities and People's Region.

intervention. In Nasarawa, four LGAs consisting of two similar pairs have been selected for evaluation. Two of these will receive the intervention (one in each pair) and two will not, that is, they will act as comparisons. In Ogun, the evaluation will be conducted in only two LGAs (one intervention and one comparison).

### Study unit inclusion criteria and selection

Study states were selected by SFH. The selected states were chosen because of the absence of other large-scale adolescent focused sexual and reproductive health activities and because of SFH's previous experience working in these states.

Study LGAs were selected by SFH in collaboration with the state Ministry of Health and local government officials. The LGAs were selected from among those where there were no security concerns, and comparison-intervention pairs were selected to be similar with respect to as many as possible of the following criteria:

► Population density.
► Estimated modern contraceptive prevalence rate (mCPR) among 15–49 year olds (DHIS2, 2016).[17]
► Number of health facilities.
► Presence of World Bank support for Maternal and Child Health activities.

### Allocation to intervention and comparison arms

Within a pair, allocation of an LGA to the intervention or comparison arm of the outcome evaluation study was done by SFH in collaboration with the state Ministry of Health and local government officials.

### Ethiopia

The intervention will be evaluated through two population-based cross-sectional surveys, one conducted before implementation (late 2017) and another approximately 24 months after the start of the intervention.

### Study unit inclusion criteria and selection

The study region and woredas were selected by PSI. Oromia region was selected because of its relatively low mCPR as compared with other regions in the Ethiopia Demographic and Health Survey (DHS) 2011 (24.9%) and its standing as having the highest unmet need for contraception (29.9%) as compared with other regions.[18] Criteria used by PSI for selecting woredas for inclusion in the study included:

► Good infrastructure and accessible all year round.
► Close proximity to PSI head office in Addis Ababa.
► Population of married adolescent girls anticipated to be large.

### Tanzania

As in Ethiopia, the intervention will be evaluated through before-and-after population-based surveys that are scheduled to take place in late 2017 and approximately 24 months after the start of the intervention.

### Study unit inclusion criteria and selection

Mwanza Region was selected by the evaluators in collaboration with PSI because of the high unmet need for modern contraception among girls aged 15–19 years relative to other A360 target regions,[9] the absence of other large-scale sexual and reproductive health activities and because PSI has previous experience working in the region. The study will be restricted to urban and semiurban wards in Ilemela District, in part because PSI focuses efforts in more densely populated areas and in part because of resource constraints.

## Study population

A360 targets different subpopulations of adolescent girls in the three countries (table 2).

### Inclusion criteria

Adolescent girls aged 15–19 years:
► Unmarried (Tanzania and Southern Nigeria only).
► Married or living as married (Ethiopia, Tanzania and Northern Nigeria only).
► Living, at the time of the survey, in the study sites.
► Voluntarily provides informed consent.

### Exclusion criteria

There were no specific exclusion criteria.

## Sampling strategy

A one-stage cluster sampling design will be used in Nigeria and Ethiopia. A two-stage cluster sampling design will be used in Tanzania. In each country, we will use the smallest available administrative unit as the primary sampling unit. Specifically, we will use enumeration areas (EAs) from the 2006 census in Nigeria, kebele from the 2007 census in Ethiopia and streets in Tanzania.

### Nigeria

The Nigeria Population Commission maintains a database of about 600 000 EAs across the entire country, which shows that EAs in Nasarawa have an average of 44 households and EAs in Ogun have an average of 52 households. They advised us that EA population counts are considered unreliable, which creates issues when using proportional to population size sampling techniques. To overcome this issue, we will use a simple random sample of EAs. To maximise the efficiency, in terms of logistics, of the study, we will sample clusters of approximately 100 households. Thus, if a selected EA contains fewer than 100 households, then we will randomly sample one of the neighbouring EAs and enumerate all households in that EA. If 100 households are still not enumerated, then a further additional EA will be randomly sampled and enumerated. We estimate that, using this technique, a simple random sample of approximately 710 EAs in Ogun and 1150 EAs in Nasarawa will need to be selected to reach the target sample size of adolescent girls. All eligible unmarried girls aged 15–19 years in the selected households in Ogun and all eligible married girls aged 15–19 years in Nasarawa will be recruited to participate in

the structured interview. In Nasarawa, in order to reach our target sample size of 250 husbands, we estimated that for every 17 sexually active married girls interviewed, one will be systematically selected and asked permission to interview her husband. In Ogun, due to the uncertainty in the proportion of unmarried girls who will report that they are sexually active, we propose that for every 7–14 sexually active unmarried girls surveyed, one will be systematically selected and asked permission to interview a cohabiting adult in order to reach our target sample size of 250 cohabiting adults. The exact sampling interval will be finalised following the pilot study.

### Ethiopia

A sample of 45 kebeles will be selected from across the four study woredas with probability proportional to population size. Within the selected kebele, we will visit each household and all eligible married girls aged 15–19 years will be recruited to participate in the structured interview. In households that have more than one eligible married female aged 15–19 years, all consenting married adolescent girls will be interviewed. In order to reach our target sample size of 128 husbands, we estimated that for every 15 sexually active married adolescent girls aged 15–19 years interviewed, one will be systematically selected and asked permission to interview her husband.

### Tanzania

A simple random sample of 30 'streets' (neighbourhoods) will be selected from across the 15 urban and semiurban wards of Ilemela District. The boundaries of each selected street will be identified and mapped using Global Positioning System (GPS) devices,[19] and within each street, we will randomly select 50 GPS coordinates using ArcGIS software V.9.3 (Esri, Redlands, USA).[20] All households whose front doors are located within a radius of 20 m around the GPS point will be visited and all eligible consenting girls aged 15–19 years residing in these households invited to be interviewed. We aim to interview 166 girls per street. In order to reach our target sample size of 127 cohabiting adults, we estimated that for every 10 sexually active adolescent girls aged 15–19 years interviewed, one will be systematically selected and asked permission to interview her husband (married girls) or a cohabiting adult (unmarried girls).

### Data collection

The questionnaires will be adapted from various research instruments that have been used and validated in the study countries, including DHSs[7–9] and FP2020 surveys.[1] They will be developed in English and then translated into the local languages of the study communities. Final modifications will be made to the questionnaires following an extensive pretesting exercise and after pilot surveys are conducted in communities outside of the selected study sites.

Questionnaires will be administered face-to-face by female interviewers aged between 18 and 26 years. For

households with potentially eligible study participants who are not at home, we will attempt to revisit up to twice (three visits in total). Data will be collected and recorded electronically in the field via tablets. This allows improved data quality through real time data delivery, built-in logical checks and skip patterns.

### Study outcomes

Our primary outcome, the mCPR among 15–19 year old girls will be defined as follows:

$$\frac{\textit{Number of fecund sexually active } 15-19 \textit{ year old girls reporting use of modern contraceptives at the time of the survey}}{\textit{Number of fecund sexually active } 15-19 \textit{ year old girls}}$$

Modern contraception: male and female sterilisation, contraceptive implants, intrauterine contraceptive devices, injectables, contraceptive pill/oral contraceptives, emergency contraceptive pill, male condom, female condom, Standard Days Method, Lactational Amenorrhoea Method, diaphragm, spermicides, foams and jelly.

Sexually active girls: those who report having sexual intercourse in the last 12 months.

Fecund girls: those who have started menstruating, are not pregnant and do not report that they are infertile.

Secondary outcomes are outlined in table 3.

### Data analysis

In Nigeria, our hypothesis is that sexually active girls aged 15–19 years living in areas where the A360 programme is implemented will have a greater increase in use of modern contraception compared with sexually active girls aged 15–19 years living in areas where the A360 programme has not been implemented. We will estimate the intervention effect separately for Ogun and each LGA-pair in Nasarawa. The effect estimates will be obtained by subtracting the baseline mCPR prevalence estimates from the prevalence estimates at follow-up and then calculating the difference between the comparison and intervention LGAs (the difference in difference method). If the two effect estimates for Nasarawa are similar, then we will combine them (eg, by weighting by the inverse of the variance) to produce a single summary estimate for the state. Similar analyses will be performed to look at the intervention effect on secondary outcomes (table 3). The difference in differences analysis assumes a common trend in the outcome in both the intervention and comparison area. Even if the two areas differ in a number of characteristics, the analysis is valid provided the common trend assumption is upheld. We will measure potential confounders at baseline and endline and adjust our analysis for any compositional changes over time in these confounders.

In Ethiopia and Tanzania, our hypothesis is that the prevalence of modern contraceptive use among sexually active girls aged 15–19 years living in areas where the A360 programme is implemented will increase between 2017 and 2019. The increase in mCPR will be greater

**Table 3** Primary and secondary outcomes

| Outcome domain | Indicators |
| --- | --- |
| Primary outcome | Prevalence of modern contraceptive use among sexually active girls aged 15–19 years |
| Secondary outcomes | ► Age-specific fertility rates.<br>► Age at first birth.<br>► Unmet need for modern contraception among sexually active girls aged 15–19 years.<br>► Adolescent girls' knowledge on the use of modern contraceptives to prevent unintended pregnancies.<br>► Adolescent girls' agency (self-efficacy) to use modern contraceptives to prevent unintended pregnancies.<br>► Adolescent girls' attitudes towards the use of modern contraceptives to prevent unintended pregnancies.<br>► Adolescent girls' access to contraceptive services and products.<br>► Adolescent girls' misconceptions about modern contraceptives.<br>► Community acceptance and social support for adolescent girls to adopt healthy sexual and reproductive health (SRH) behaviours, including use of modern contraceptives. |

than what would have been expected to have occurred in the absence of the intervention. The primary analysis will compare the proportion of sexually active girls who report using modern contraception at baseline and endline. Similar analyses will be performed to look at the intervention effect on secondary outcomes (table 3). We will use logistic regression models to adjust for potential confounders, including age, educational attainment, parity and marital status (Tanzania only). In addition, we will conduct the following secondary analyses:

1. Dose–response: individual level of engagement with the A360 interventions will be measured at endline. We will use a series of questions to rank individuals by their level of engagement with the A360 interventions that are available in the place where they live. We will then use regression models to assess the strength of association between level of engagement with the A360 interventions and use of modern contraception. Our hypothesis will be that those who are more engaged will be more likely to use modern contraception. If the data are consistent with this hypothesis, this will provide additional evidence of the effectiveness of the intervention. Our analysis approach will try to capture exposure to the main components of the intervention and to capture overall exposure to the package of interventions. For example, if there are two main components of the intervention (A, B), then we may have three different exposure variables (A, B and a combination of A and B). There remains some uncertainty as to which intervention components will be implemented in each setting over the 2-year follow-up. A detailed analysis plan will be finalised prior to the endline data collection.

2. Secular trends: mCPR data available from other sources for the time period 2015–2019 will be examined to assess whether changes in mCPR in A360 communities between 2017 and 2019 reflect background changes in mCPR or whether mCPR appears to have increased more than would be expected during this time period. Detailed contraceptive use data among adolescents

do not exist for the specific geographical areas in our study. Potential sources of data include DHS (Tanzania, Ethiopia),[8 9] PMA2020 (Ethiopia),[21] and demographic surveillance site data from areas near to our study sites (Kisesa in Mwanza[22]; Kersa and Harar in Oromia[23]). These data are unlikely to be directly comparable but we believe that they will give a broad indication as to whether mCPR is increasing, static or decreasing in the regions our study are situated in. Specifically, we will undertake a 'modified' difference in difference analysis. Prior to analysis of the endline data, from available existing data we will estimate the absolute increase in mCPR rates that we would expect in the study communities if secular trends from other sources were replicated. In the 'modified' difference in difference analysis, we will subtract this expected change from the difference between endline and baseline mCPR rate. When presenting this 'modified' difference in difference analysis, we will also present our assessment of the likely comparability, accuracy and completeness of these alternative data sources.

All analyses will be conducted in Stata V.15 and we will use weights and robust SEs to account for the cluster sampling design.

### Sample size

The baseline mCPR estimates and projected trends used in our sample size calculations were based on analysis which PSI conducted in September 2015 using available DHS data and our own review of historical DHS and PMA2020 data.[7 18 24 25]

Effect estimates are based on an analysis (unpublished) conducted by one of our evaluation collaborators, Ms Michelle Weinberger (Avenir Health). She reviewed 25 studies published between 1993 and 2014, which had estimated the impact of family planning interventions on mCPR using a variety of study designs.[26–44] Ms Weinberger extracted published ORs for effect size when available or calculated them using published results. She then calculated the median and maximum ORs to give a sense of

**Table 4** Assumptions for key parameters required for sample size calculations

| Parameter | Ogun* (%) | Nasarawa* (%) | Oromia† (%) | Mwanza‡ (%) |
|---|---|---|---|---|
| Proportion of 15–19-year-old girls who are married (or living together) | 10 | 15 | 20 | 22 |
| Proportion of 15–19-year-old girls who are unmarried (not currently married) | 90 | 85 | 80 | 78 |
| Proportion of unmarried 15–19-year-old girls who report sexual activity in the past year | 15 | 15 | – | 25 |
| Proportion of married 15–19 year olds who report sexual activity in the past year | 97 | 97 | 97 | 97 |
| Proportion of sexually active girls who are married | 42 | 53 | – | 52 |
| Proportion of sexually active girls who are unmarried | 58 | 47 | – | 48 |
| Proportion of households with resident who is female aged 15–19 years | 19 | 29 | 27 | 34 |

Sources:
*Nigeria DHS 2013,[7] Nigeria GHS 2016.[13]
†Ethiopia DHS 2011,[18] Ethiopia Mini DHS 2014,[24] Ethiopia census 2007.[14]
‡Tanzania DHS 2015–2016,[9] Tanzania census 2012.[15]
DHS, Demographic and Health Survey.

the average and maximum increase in mCPR expected based on the existing evidence base.

### Nigeria

In Ogun State, among sexually active unmarried girls aged 15–19 years, we assumed that between 2017 and 2019, mCPR will increase from 64.4% to 65.6% in the absence of A360 and from 64.4% to 72.6% in the presence of A360. Based on these assumptions and those in tables 4 and 5, our target sample size is 12 020 unmarried girls aged 15–19 years.

In Nasarawa State, among sexually active married 15–19 year olds, we have assumed that between 2017 and 2019, mCPR will increase from 3.0% to 3.1% in the absence of A360 and from 3.0% to 5.1% in the presence of A360. Thus, 4555 married girls aged 15–19 years must be surveyed to achieve 90% power (table 5).

In addition, a sample of 250 cohabiting adults (Ogun) and 250 husbands/male partners (Nasarawa) will be interviewed.

### Ethiopia

In Oromia Region, among sexually active married girls aged 15–19 years, we have assumed that between 2017 and 2019, mCPR will increase from 44.0% to 50.8% in the presence of A360. Based on this assumption and those in table 4, 1926 married girls aged 15–19 must

be surveyed to achieve 90% power (table 5). In addition, a sample of 128 husbands/male partners will be interviewed.

### Tanzania

In Ilemela District, Tanzania, among sexually active girls aged 15–19 years, we have assumed that between 2017 and 2019 mCPR will increase from 26.7% to 32.7% in the presence of A360. Thus, 4980 girls aged 15–19 years (corresponding to 1217 sexually active girls) must be surveyed to achieve 90% power (table 5). In addition, a sample of 127 cohabiting adults will be interviewed.

Based on discussions between the A360 funders, implementers and ourselves as independent evaluators of the programme, it was felt to be important to have an evaluation which was powered to detect small increases in mCPR in the study settings. These small effect sizes would be important in terms of the number of users of contraception gained and potential unplanned pregnancies averted among adolescent girls given the large scale of rollout of A360 across the three countries. In addition, it was felt that realistically, over only 2 years, achievable effect sizes were likely to be small but if detected might provide some reassurance that in the longer term we could expect greater increases.

**Table 5** Sample size estimates

| Parameter | Ogun | Nasarawa | Oromia | Mwanza |
|---|---|---|---|---|
| Target sample of sexually active 15–19 year olds* | 1413 | 3586 | 1132 | 1217 |
| Total sample of all 15–19 year olds*<br>► Includes non-sexually active girls<br>► Taking into account 10% non-response | 10 362 | 4067 | 1284 | 3314 |
| Design effect | 1.16 | 1.12 | 1.50 | 1.50 |
| Sample size for survey† | 12 020 | 4555 | 1926 | 4980 |

*Includes unmarried girls (Ogun); married girls (Nasarawa and Ethiopia); unmarried and married girls (Mwanza).
†Total sample of all 15–19 year olds × design effect.

Further details of the sample size calculations for the study, including sources of data used for assumptions, are available in the online supplementary file 1 .

## Patient and public involvement

There was no patient or public involvement in the design of this study. However, the intervention was designed using a human-centred design process which includes an iterative process of generating, testing and refining ideas and developing and testing prototypes with individuals from the target population through a series of structured workshops.[16]

## Strengths and limitations

A strength of our outcome evaluation is the collection of comparable data before and after intervention implementation in four different settings in three countries. In two settings in Nigeria, we will also collect data from populations not exposed to the intervention and hence will have a quasi-experimental design.[45] Triangulation with dose–response and trends analyses and implementer monitoring and evaluation data will strengthen the inference possible from the study findings. A process evaluation, conducted throughout the 2-year implementation period in the outcome evaluation areas, will provide information on the context and mechanism of the intervention and will complement the outcome data.

In Nigeria, where the intervention will be evaluated through a quasi-experimental design, the validity of the effect estimate depends on the time trend being the same in both intervention and comparison areas. The common trend assumption is untestable because we will not know what the trend in the intervention area would have been in the absence of the intervention. We have tried to select LGAs with similar key sociodemographic and reproductive health indicators as trends in mCPR are likely to be influenced by the characteristics of the population. In addition, our baseline survey will allow us to undertake a more accurate assessment as to how comparable our study sites are on a number of additional key indicators and we will be able to adjust for imbalances in potential confounders at the analysis stage.

In Tanzania and Ethiopia, the study design does not include a comparison group and observed changes in mCPR could be due to secular trends or other influences. As described above, we will examine historical and contemporaneous mCPR data from other sources so that our findings can be interpreted in the context of underlying trends. In addition, a dose–response analysis will be conducted at endline to look at the association between individual-level engagement with the A360 intervention and modern contraception use.

Due to resource constraints, we decided to focus on a limited number of geographical areas, which will affect the generalisability of our findings. However, our study is only one component of the overall A360 evaluation. The A360 programme implementers will also be collecting monitoring and evaluation data across all sites, and the process evaluation and a cost effectiveness analysis will be conducted over

wider A360 areas. We anticipate incorporating this additional information into the overall evaluation.

It is important to note that this outcome evaluation is not evaluating human-centred design per se, but an intervention designed using human-centred design. A major challenge in designing the outcome evaluation for A360 was that when the outcome evaluation study protocols and data collection tools were being developed, the A360 project was in the mid-stages of intervention development. If the final intervention package is significantly different from earlier prototypes, then the study protocol and data collection tools for the baseline study may not be as well tailored to the intervention as if the final package of interventions was known in advance. If needed, changes to the endline study protocol and data collection tools will be made to better capture the impact of the final A360 package of interventions.

## ETHICS AND DISSEMINATION
### Informed consent

Informed consent will be obtained from all study participants and the consent process will be documented. Written informed consent will be obtained from all participants in Tanzania. In Nigeria and Ethiopia, only verbal consent will be obtained as a waiver of written consent was granted because national surveys obtaining verbal responses to a questionnaire and involving similar sensitive questions on reproductive health issues are carried out using only verbal consent in these settings. In Nigeria, parental consent and adolescent girl assent are required for unmarried girls aged 15–17 years. In Tanzania, parental consent waiver was granted for this age group because of the sensitive nature of the topics discussed. Married adolescent girls under 18 years of age are considered emancipated in all three countries and do not require parental consent in addition to their own voluntary consent. Study participants will be informed of all risks and protections and will be able to withdraw from the study at any time for any reason.

### Benefits and risks

There are no direct individual benefits for taking part in the study. Potential risks to participants are minimal. The most significant risk identified is a breach of confidentiality. There will be no identification of the respondent on the survey questionnaire. All study staff will be trained to ensure that they will protect the privacy and confidentiality of participants to the fullest extent possible. Interviews will be held at the household of the respondents out of hearing range of others. All data will be entered directly into tablets and sent to private secure servers on a daily basis through a private and secure internet connection. Data security will include data storage encryption and controlled password protected access for authorised users only. Data will be kept anonymised during the study and will be kept strictly confidential in storage for 10 years after completion of the study. All data based on this

research will be reported in aggregate form. Participants will not be identified by name in any report or publication resulting from the study data.

## Compensation

There are no costs for being in the study. Therefore, participants will not receive compensation for taking part.

## Dissemination of study findings

Our research findings dissemination plan includes peer-reviewed publications, stakeholder workshops, reports and briefings, social media and presentations at different forums. In compliance with the funder's requirements, after a period of 6–12 months following the endline survey, the data will be made available via the London School of Hygiene and Tropical Medicine Data Repository after removing all direct and indirect identifiers from the data.

**Author affiliations**
[1]MRC Tropical Epidemiology Group, London School of Hygiene and Tropical Medicine, London, UK
[2]Itad Limited, Hove, UK
[3]Department of Infectious Disease Epidemiology, London School of Hygiene and Tropical Medicine, London, UK
[4]Mwanza Intervention Trials Unit, Mwanza, Tanzania
[5]Binomial Optimus Limited, Abuja, Nigeria
[6]MMA Development Consultancy, Addis Ababa, Ethiopia
[7]Department of Social and Environmental Health Research, London School of Hygiene and Tropical Medicine, London, UK

**Acknowledgements** Ms Michelle Weinberger (Avenir Health) for providing the effect estimates for our sample size calculations. Itad as the lead organisation responsible for the overall A360 evaluation. PSI Headquarters, PSI Ethiopia, PSI Tanzania and SFH for their support with site selection and engagement in conversation regarding the design. Contributions from London School of Hygiene and Tropical Medicine authors are part of their work for the Centre for Evaluation, which aims to improve the design and conduct of public health evaluations through the development, application and dissemination of rigorous methods and to facilitate the use of robust evidence to inform policy and practice decisions.

**Contributors** CJA, EM, SK, MKN, EEC, MM, CB, JRH and AMD were involved in conception and study design. CB provided statistical expertise. CJA, EM and AMD were involved in drafting of the manuscript. SK, MKN, EEC, MM, CB and JRH were involved in critical revision of the manuscript for important intellectual content. All the authors were involved in final approval of the manuscript and decision to submit the manuscript for publication.

**Funding** The Bill & Melinda Gates Foundation and the Children's Investment Fund Foundation.

**Competing interests** None declared.

**Patient consent** Not required.

**Ethics approval** National Health Research Ethics Committee of Nigeria (Ref: NHREC/01/01/2007-25/05/2017), National Health Research Ethics Review sub-Committee of Tanzania (Ref: NIMR/HQ/R.8a/Vol. IX/2549), Oromia Health Bureau Research Ethical Review Committee (Ref: BEFOIHBTFH/1-8/2844) and the London School of Hygiene and Tropical Medicine Ethics Committee (Ref: 14145).

**Provenance and peer review** Not commissioned; externally peer reviewed.

**Data sharing statement** This is a study protocol paper. No data are yet available. Our research findings dissemination plan includes peer-reviewed publications, stakeholder workshops, reports and briefings, social media and presentations at different forums. In compliance with the funder's requirements, after a period of 6–12 months following the endline survey, the data will be made available via the London School of Hygiene and Tropical Medicine Data Repository after removing all direct and indirect identifiers from the data.

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
