## [Reviewer comments · BMJ Open]

ARTICLE DETAILS

TITLE (PROVISIONAL)	Evaluating the impact of an intervention to increase uptake of modern contraceptives among adolescent girls (15 to 19 years) in Nigeria, Ethiopia and Tanzania: the Adolescents 360 quasi-experimental study protocol
AUTHORS	Atchison, Christina; Mulhern, Emma; Kapiga, Saidi; Nsanya, Mussa Kelvin; Crawford, Emily; Mussa, Mohammed; Bottomley, Christian; Hargreaves, James; Doyle, Aoife

VERSION 1 – REVIEW

REVIEWER	Ellen Løkkegaard Department of Obstetrics and Gynecology North Zealand Hospital, Hillerød University of Copenhagen Denmark
REVIEW RETURNED	19-Feb-2018

GENERAL COMMENTS	This is a very well written protocol paper and look forward to the results from the study on the important issue of intervention to increase uptake of modern contraceptives among adolescent girls in Nigeria, Ethiopia and Tanzania. I have a few comments. First the use of abbreviations. I would improve the readability of the paper to not use abbreviations. As the methods for the intervention cannot be described in detail due to the nature of the study I have had to say no to whether the study is described sufficiently so the study can be repeated. It is not clear to me how the power calculation was performed, that is why I suggest statistical review. It would be informative if the references for Ms Michelle Weinberges unpublished review (Page 14 line 12) were included in the reference list.
---

REVIEWER	Lisa Dulli, Scientist II FHI 360, USA
REVIEW RETURNED	28-Feb-2018

GENERAL COMMENTS	In attempting to combine information from three separate impact evaluations of related, but different intervention strategies in three different countries, that use differing study designs, it seems as though much of the detail necessary to a research protocol is left out. There is insufficient information provided for each of the three studies to thoroughly convey the necessary details for the work. I have some specific concerns about the methods, particularly with regard to sampling. Although there is mention of a review of 25
---

	studies to inform effect size estimation, there are no references for those studies. The large sample sizes that result from a choice of 90% power and very small effect sizes will result in large resource expenditures. No argument is made as to why it is important to be able to detect such small effect sizes. The authors also state that they are using a two-stage cluster sampling approach, but their description is a single stage cluster sampling approach. There is no mention as to how the design effects were calculated. There is no explanation for the sample sizes for men in the study. There is a brief description of the process evaluation component of the study that introduces new data collection and study populations, but only in cursory detail. The section on ethical considerations is lacking. - The reviewer also provided a marked copy with additional comments. Please contact the publisher for full details.
--	---

VERSION 1 – AUTHOR RESPONSE

Response to Reviewer 1 comments

- 1. First the use of abbreviations. I would improve the readability of the paper to not use abbreviations.**

Standard and well recognised abbreviations have been used for our primary outcome (prevalence of voluntary use of modern contraception - mCPR) and other terms that are internationally commonly used abbreviations (e.g. Demographic Health Survey - DHS). All abbreviations in the manuscript are used only after they are fully spelt out on first appearance in the manuscript. However, we acknowledge the reviewers concerns and have removed the abbreviations which are not commonly used in the published literature.

- 2. As the methods for the intervention cannot be described in detail due to the nature of the study I have had to say no to whether the study is described sufficiently so the study can be repeated. It is not clear to me how the power calculation was performed, that is why I suggest statistical review.**

We have included an online supplementary file with further details on sample size calculations.

- 3. It would be informative if the references for Ms Michelle Weinberger's unpublished review (Page 14 line 12) were included in the reference list.**

References have been given for Michelle Weinberger's unpublished review. We have been informed by Michelle that this review is being prepared for publication.

Response to Reviewer 2 comments

Reviewer 2's general comments are acknowledged and responded to specifically by addressing each individual comment in the pdf file provided to us by the Editor.

- 1. Nothing in the introduction to support the intervention approach(es) under study.**

We have included an additional paragraph in the introduction summarising the literature on challenges faced by adolescent girls accessing contraception and effective interventions in adolescent sexual and reproductive health programming.

- 2. Comparison**

We have changed the wording (highlighted in yellow) from control to comparison arm where this appears in the manuscript.

3. For a protocol, it would be appropriate to specify which criteria, not leave it open.

It was not possible to match on all the criteria when selecting comparison-intervention pairs. We attempted to match on as many as possible. The wording (highlighted in yellow) in section “Design of outcome evaluation” has been changed to make this clearer.

4. Exclusion criteria are not typically just the opposite of the inclusion criteria, but rather conditions for which those who otherwise meet the inclusion criteria would be excluded.

We accept this point. We have removed the exclusion criteria and stated that there were no specific exclusion criteria. Wording (highlighted in yellow) in section “Study population” has been changed.

5. In each of the countries? If so, specify.

We have added additional text (highlighted in yellow) on consent to the “Ethics and Dissemination” section to address this comment.

6. All of these that you describe constitute a single stage cluster sampling approach, not two stage. To be a two stage approach, you would first sample your PSU, then you would sample secondary units from within the PSU. What you describe is taking all units within the PSU, thus no sampling is done at that level.

We accept this point for Ethiopia and Nigeria. Wording (highlighted in yellow) in Table 2 and section “Sampling strategy” has been changed to make clear this was a single stage cluster design. However, in Tanzania we have a two stage design. First, we sample our PSU (street) and then we sample secondary units (area defined by GPS point and 20m radius) within the streets.

7. All eligible individuals to be *recruited* to participate.

Wording (highlighted in yellow) in Table 2 has been changed as reviewer suggests.

8. Why call this a probability sample, as opposed to a SRS like above?

In Ethiopia we did not do SRS but sampled kebele with probability proportional to their population size (PPS). We used a slightly different sampling approach in Ethiopia than in Nigeria and Tanzania as we had the population sizes of the kebele. This will be adjusted for in the analysis of the Ethiopia data. Wording (highlighted in yellow) in Table 2 and section “Sampling strategy” has been changed to make this clear.

9. This is inconsistent with the table. Why "clusters" of 100 households? How are these clusters defined?

We have added additional text (highlighted in yellow) to clarify the sampling strategy with respect to EAs in Nigeria in the section “Sampling strategy”.

10. All eligible girls will be recruited to participate in the structured interview - maintain voluntary nature of the research.

Wording (highlighted in yellow) in section “Sampling strategy” has been changed as reviewer suggests.

11. On what is the number of male partners based?

Please see supplementary file (Table S4) for further details on sample size calculations.

12. Simple random sample for consistency.

In Ethiopia, we did not do SRS but sampled kebele with probability proportional to their population size (PPS). We used a slightly different sampling approach in Ethiopia than in Nigeria and Tanzania as we had the population sizes of the kebele. This will be adjusted for in the analysis of the Ethiopia data. Wording (highlighted in yellow) in Table 2 and section “Sampling strategy” has been changed to make this clear.

13. recruit all to participate

Wording (highlighted in yellow) in section “Sampling strategy” has been changed as reviewer suggests.

14. Why not use enumeration areas.

As stated at the beginning of the section “Sampling strategy”, in each country we will use the smallest available administrative unit as the primary sampling unit (PSU). In Mwanza the smallest available administrative unit was the “street”.

15. Why do you only mention the number of times a household will be approached in this section?

16. Can delete the similar information for Tanzania above.

This sentence has been deleted as suggested in the later comment by the reviewer.

17. How will you analyze these secondary outcomes? Will you be comparing them statistically or simply describing them?

We will be comparing them statistically in a similar way as for our primary outcome. We have included additional text (highlighted in yellow) in the “Data Analysis” section to make this explicit.

18. Because you state you are studying effectiveness of an intervention, you should state the hypothesis that you are testing.

We have included additional text (highlighted in yellow) in the “Data Analysis” section to state our hypothesis.

19. Marital status cannot be a control variable if it is an eligibility criterion.

In Tanzania our study population is both married and unmarried women. We have included additional text (highlighted in yellow) in the “Data Analysis” section to state that marital status will only be considered as a potential confounder in Tanzania.

20. On the face of it, this might make sense, but your interventions are multi-faceted and some components of the interventions may have a greater potential impact on the outcome than other components. How will you control for those differential effects?

Yes, this will be complex and there remains some uncertainty as to which intervention components will be implemented in each setting over the two year follow-up. Our analysis approach will try to capture exposure to the main components of the intervention and to capture overall exposure to the package of interventions. For example, if there are two main components of the intervention (A, B) then we may have three different exposure variables (A, B and a combination of A & B). A detailed analysis plan will be finalised prior to the endline data collection. We have included additional text (highlighted in yellow) in the “Data Analysis” section to address the reviewer’s comment.

21. It will be very important to ascertain the degree to which these measures are or are not comparable with your measures. For example, facility-based service statistics represent a different population than population-based surveys like the DHS.

We agree and will take this into account. We have included additional text (highlighted in yellow) in the “Data Analysis” section to make this explicit.

22. If accurate existing data are available, then why do you need to collect your own data? How comparable are these data, how accurate are they and how reliable/complete are they?

Detailed contraceptive use data among adolescents do not exist for the specific geographical areas that we are working in. The existing data that we have in mind are: DHS data, PMA2020 data and data from demographic surveillance sites which are near to our study sites e.g. Kisesa in Mwanza (Tanzania) and Kersa and Harar in Oromia (Ethiopia). These data are unlikely to be directly comparable but we believe that they will give a broad indication as to whether mCPR is increasing, static or decreasing in the regions that the study are situated in. When presenting this modified Difference-in-Difference analysis, we will also present our assessment of the likely comparability, accuracy and completeness of these alternative data sources. We have included additional text (highlighted in yellow) in the “Data Analysis” section to make this explicit.

23. Why the choice of 90% power? How were your within cluster estimates of variability determined?

Typical values for power are 80%, 90% and 95%. There is always a compromise between power and sample size because raising *power* will require increasing the *sample size*. However, 90% power is commonly used in sample size calculations (e.g. it is used in the established medical statistics textbook “Essentials of Medical Statistics” by Kirkwood and Sterne).

Further details on the intra-cluster correlation coefficients used are provided in the supplementary online file (Table S3).

24. Cite

References have been given for Michelle Weinberger’s unpublished review. We have been informed by Michelle that this review is being prepared for publication.

25. Cite source of assumption.

The mCPR estimates and projected trends used in our sample size calculations were based on analysis which PSI conducted in September 2015 using available DHS data and on our own review of historical DHS and PMA2020 data. Please see supplementary file for further details. This will be submitted as an online supplementary file to the main protocol manuscript.

26. So in Nigeria are you planning a difference in differences analysis?

Yes, please see text in “Data Analysis” section which explicitly states we will be doing a difference in differences analysis in Nigeria.

27. Is this small effect size worth the effort?

Based on discussions between the A360 funders, implementers and ourselves as independent evaluators of the programme, it was felt to be important to have an evaluation which was powered to

detect small increases in mCPR in the study settings. These small effect sizes would be important in terms of the number of users of contraception gained and potential unplanned pregnancies averted among adolescent girls given the large scale of roll out of A360 across the three countries. In addition, it was felt that realistically, over only two years, achievable effect sizes were likely to be small but if detected might provide some reassurance that in the longer term we could expect greater increases. We have included additional text (highlighted in yellow) in the “Sample size” section to make this justification.

28. Why 90% power? Is this effect size meaningful? Can you argue the intervention is worth the investment if you can only increase mCPR by 2.1 percentage points?

Please see responses to Comments 23 and 27.

29. What is the basis of the sample size calculations for men?

Please see supplementary file (Table S4) for further details on sample size calculations. This will be submitted as an online supplementary file to the main protocol manuscript.

30. Same issues with regard to power, effect size and sampling of men as in Nigeria (and in Tanzania).

Please see responses to Comments 23, 27 and 29.

31. How was this calculated? What estimate of within cluster variation was used?

Further details on the intra-cluster correlation coefficients used are provided in the supplementary online file (Table S3).

32. The process evaluation is part of your research protocol so all of these data collection methods need to be expanded to cover the methods in detail. What study objective do these IDIs address? What is the purpose of the FGDs? What study objective do they address and how? How will participants be selected? What data will be collected?

We have removed the section on the Process Evaluation. As the reviewer indicates, a lot of the detail is lacking. This is primarily due to the fact that the Outcome Evaluation requires a lot of detailed description. The Process Evaluation protocol was developed separately and will now be written up for publication and described in detail separately elsewhere. We now state in the introduction section that the Outcome Evaluation described in our manuscript is part of a multi-component evaluation including a process evaluation and cost-effectiveness study.

33. What will you do if they are not comparable?

The difference in differences analysis assumes a common trend in the outcome in both the intervention and comparison area. Even if the two areas differ in a number of characteristics, the analysis is valid provided the common trend assumption is upheld. We will measure potential confounders at baseline and endline and adjust our analysis for any compositional changes over time in these confounders. We have included the additional text (highlighted in yellow) in the “Data analysis” section.

34. Or other influences.

We have included the additional text (highlighted in yellow) in the “Strengths and Limitations” section.

35. This is missing extensive details on ethical considerations - informed consent, compensation, risk/benefits, etc.

We have added additional text (highlighted in yellow) to the “Ethics and Dissemination” section to address this comment.